# Numerical Investigation of the Elastic Properties of Polypropylene/Ultra High Molecular Weight Polyethylene Fiber inside a Composite Material Based on Its Aspect Ratio and Volume Fraction

**DOI:** 10.3390/polym14224851

**Published:** 2022-11-10

**Authors:** Jong-Hwan Yun, Yu-Jae Jeon, Min-Soo Kang

**Affiliations:** 1Mobility Materials-Parts-Equipment Center, Kongju National University, Gongju-si 32588, Korea; 2Department of Medical Rehabilitation Science, Yeoju Institute of Technology, Yeoju 12652, Korea; 3Division of Smart Automotive Engineering, Sun Moon University, Asan-si 31460, Korea

**Keywords:** polypropylene, ultrahigh molecular weight polyethylene, composite, aspect ratio, volume fraction

## Abstract

In this study, the characteristics of a composite material composed of polypropylene (PP) and ultrahigh molecular weight polyethylene (UHMWPE) are investigated. The elastic properties of the PP/UHMWPE composite material composed of short UHMWPE fibers with a low aspect ratio and long UHMWPE fibers with a high aspect ratio are compared and analyzed. In addition, the elastic properties of the PP/UHMWPE composite materials are calculated via finite element analysis and the Halpin–Tsai model by changing the volume fraction of the UHMWPE fibers. The results show that when UHMWPE fibers with a low aspect ratio and volume fraction are used, the results of the modulus of elasticity based on the finite element analysis are consistent with those obtained using the Halpin–Tsai model, although the fiber volume fraction of the UHMWPE fibers increases. Meanwhile, the deviation between the results yielded by both methods increases with the aspect ratio of the fiber. In terms of the shear modulus, the Halpin–Tsai model shows a linear trend. The results from the finite element analysis differ significantly from those of the Halpin–Tsai model owing to the random orientation of the UHMWPE fibers inside the fiber. Using a contour graph constructed based on the finite element analysis results, the aspect ratio and volume fraction of the UHMWPE fibers can be inversely calculated based on the elastic properties when synthesizing a PP/UHMWPE fiber composite. In future studies, the interfacial bonding properties of UHMWPE fibers and PP should be compared and analyzed experimentally.

## 1. Introduction

Reducing the weight of materials is important for achieving efficient energy transportation. Recently, interest in reducing the weight of materials and components has increased owing to emission reduction regulations and the development of energy-efficient strategies. In general, metal constitutes more than 63% of the weight of a vehicle [1]. A 10% reduction in the weight of a vehicle can increase the fuel efficiency of the internal combustion engines and electric vehicles by 6–8% and 10%, respectively [2]. Hence, over the past few decades, the replacement of materials such as metal, wood, and ceramics with plastics has been actively pursued. Plastics have received significant attention owing to their ease of processing and advantages in terms of productivity and cost reduction; however, their use is limited by their low mechanical properties. Hence, researchers have attempted to develop new materials by aligning fiber reinforcing agents and reinforcing agents inside plastics to form a composite material. Consequently, fiber-reinforced polymers have been utilized in the automotive industry for several decades, thus further progressing the composites industry. The value of the global automotive composites market was estimated to be approximately $9.4 billion in 2020 and is expected to reach approximately $19.4 billion by 2027, with a composite annual growth rate of 10.9%. The application of composite materials, particularly carbon fiber-reinforced plastics and glass-fiber-reinforced fibers in automotive structures, has expanded rapidly in recent years [3,4,5,6,7,8,9,10]. However, carbon fibers are disadvantageous as they are expensive and generate harmful substances when mixed to synthesize a composite material. In addition, glass fiber-reinforced composites exhibit a low modulus of elasticity and lower strength compared with carbon fibers. Nevertheless, many studies have been conducted on these fiber-reinforced composite materials due to their excellent mechanical properties [11,12,13]. However, the types of fibers used in composite materials are limited, and research is needed to develop and use new types of reinforcing agents. Hence, the use of ultrahigh molecular weight polyethylene (UHMWPE) is emerging. UHMWPE has increased recently because of its superior wear resistance, low-temperature toughness, and tensile strength, owing to its higher molecular weight. UHMWPE offers the highest wear resistance and impact resistance among the existing polymers. Hence, it is often used to fabricate bulletproof vests, military composite helmets, fishing lines, the bottom surface of skis, biomaterials, etc. However, as UHMWPE has a relatively low melting point, its mechanical properties deteriorate rapidly as the temperature increases. Therefore, instead of solely using the material, incorporating it into another material to form a composite material is more advantageous. When developing composite materials, fillers (reinforcing agents) such as carbon and glass fibers are important; additionally, the material used for the matrix is equally important. Although various materials have been used to construct the matrix, polypropylene (PP) is the most frequently used material [14,15,16,17]. PP is a thermoplastic polymer material obtained via the chain growth polymerization of a propylene monomer comprising three carbon atoms. Similar to polyethylene, it is a representative polyolefin-based polymer and is used in various applications such as packaging, fibers, films, automobile components, storage containers, and medical products. The density of PP is lower than that of polyethylene (0.895–0.920 g/cm^3^) and is the lowest among those of general-purpose polymer materials. It is mechanically strong, yet flexible. Additionally, it is used for fabricating plastic hinges owing to its excellent robustness against repeated bending. When PP is polymerized with ethylene, its mechanical properties are further improved and it reaches a level comparable to those of high-strength plastic materials such as ABS. Therefore, a composite material developed using PP as a matrix and UHMWPE fiber as a reinforcing material can be assumed to be an effective composite material. Fiber-reinforcement mechanisms can be categorized into short-fiber [18,19,20], long-fiber [21], and continuous-fiber reinforcement mechanisms [22,23,24]. Composite materials using short fibers are advantageous for processing components as they allow for injection molding processes; however, their mechanical properties are lower than those of composite materials comprising long fibers. Nonetheless, the latter material is disadvantageous in terms of processability and formability. Hence, the lengths of the matrix and fibers must be determined by considering the formability and mechanical strength required to fulfill the purpose of the composite material to be synthesized. Factors affecting the elastic properties and mechanical strength of the composite material are associated with the directionality of the additive added as a reinforcing agent, as well as the volume fraction of fibers in the composite material. If the volume fraction of the reinforced fiber is increased, then the specific gravity that complements the mechanical strength of the composite material increases; consequently, the composite material exhibits an excellent mechanical strength. Therefore, the volume fraction of the composite materials and the length of fibers must be considered when comparing and analyzing the effects of composite materials. However, researchers have been conducting research on isotropic materials for the injection molding of PP/UHMWPE composites [25,26]. In addition, Thomas Unger et al. conducted a study on the rheological and mechanical behavior of PP/UHMWPE mixtures [27] However, until now, research regarding the design of the PP/UHMWPE fiber has been insufficient. In this situation, it is necessary to conduct analytical research to develop composite materials that can effectively utilize the excellent mechanical properties of the UHMWPE fiber. In order to effectively design a composite material, it is necessary to be able to predict the effect when each material is mixed and when the composite material is combined. As the volume fraction and aspect ratio of the fiber are factors that greatly affect the change in the elastic properties of the composite material, it is necessary to perform an analysis. In this study, the elastic properties of the composite materials synthesized using PP for the matrix and UHMWPE for reinforcement were predicted. The elastic properties of composite materials depend on the volume fractions of the matrix and reinforcement, and they change depending on the aspect ratio of the reinforcement. To analyze these characteristics, the volume fraction and aspect ratio of UHMWPE are set as variables. In this study, finite element analysis was performed by varying the volume fraction percentage (VF%) of UHMWPE to 5%, 10%, 15%, 20%, and 25%, and finite element analysis was performed by varying the aspect ratio from 3 to 15.

## 2. Materials and Methods

Figure 1 shows the tensor notation used in this study. As the fibers are evenly oriented in the matrix, a tensor is used to indicate the direction in which stress occurs so as to numerically analyze the composite material, as shown in Figure 1. When an elastic modulus is applied in the longitudinal direction of the fiber in the matrix, it is indicated as E, and the *x*-, *y*-, and *z*-axes are indicated as 1, 2, and 3, respectively, depending on the axis of the matrix. Meanwhile, the shear modulus occurring on each side of the material is denoted by G, and based on tensor notation, G_12_ implies that the shear stress occurring in the *y*-axis direction is from the *x*-axis side. This tensor notation can be applied equally to the fiber, and the length-to-diameter ratio of the fiber is defined as the aspect ratio. In addition, superscripts m and f are used for the tensor notation of the PP matrix and UHMWPE fiber, respectively, for clarity and brevity.

In this study, we calculated the elastic behavior of the composite material mixed with the PP used as the matrix and the UHMWPE fiber used as the reinforcing agent. A micromechanics model was used to predict the theoretical elastic properties of the composite materials; subsequently, the predicted results were compared with the results of the finite element analysis. Micromechanics analysis refers to an analysis based on the level of individual components that constitute complex or heterogeneous substances; it allows for the prediction of multiaxial properties that are difficult to measure experimentally. The main advantage of micromechanics analysis is that it allows for virtual tests to be performed, thus reducing the cost of physical experiments. In fact, performing experiments on heterogeneous materials is costly and involves more variables, such as the constituent material combinations, fiber and particle volume fractions, fiber and particle arrangements, and processing history. In micromechanics analysis, these parameters can be used to simulate the properties of composite materials via virtual tests. Therefore, various micromechanics models such as the rule of mixture (ROM) [28], modified rule of mixture [29,30], and Chamis model have been developed and used. Among them, the Halpin–Tsai model is a mathematical model for predicting the elasticity of composite materials based on the geometry and orientation of the filler and the elastic properties of the filler and matrix [31,32,33,34]. This model is derived based on an empirical formula obtained using a consistent experimental method [35,36,37,38]. The Halpin–Tsai model predicts the elastic properties of composite materials using empirical formulas involving the aspect ratio of the added fibers, which is based on the reinforcement factor. Therefore, to calculate the elastic properties of the composite material based on the aspect ratio and the VF% of UHMWPE in the PP matrix, theoretical simulations were performed using the Halpin–Tsai model in this study. As shown in Equations (1)–(9) [39,40], this model uses the experimentally calculated reinforcing factor (ξ) to improve the existing ROM. The reinforcement factor depends on the shape of the fibers and their arrangement, as well as the loads. In the Halpin–Tsai model, the longitudinal elastic modulus of the fiber reinforcing agent of the composite material can be obtained through Equation (1), and the vertical elastic modulus of the length of the fiber reinforcing agent of the composite material can be obtained through Equation (2). In addition, the number of preliminary steps of the composite material can be obtained through Equation (3). To obtain the elastic properties of the composite material, it can be accurately predicted through the refocusing factor of Equations (4)–(5) and the correction coefficient (η) of Equations (7)–(9). In addition, the physical properties of the PP and UHMWPE used in this study are listed in Table 1. The measured physical properties were used based on commercially available products. PP was obtained by citing Adstif EA5074 of PolyMirae Corporation, and UHMWPE used the physical characteristics of Mitsui Chemical’s MIILON XM-220 product. Although the properties of the materials used were characterized by linear isotropic materials, these isotropic properties can be applied as anisotropic materials depending on the shape of cylindrical UHMWPE fibers, modeled through ANSYS Material Designer. Accordingly, in the FEA analysis, physical properties may appear as the properties of anisotropic materials according to the modeling shape.
(1)$$E_{11}=E_{m}\frac{\left(1+\xi_{11}\eta_{11}V_{f}\right)}{\left(1-\eta_{11}V_{f}\right)}$$
(2)$$E_{22}=E_{m}\frac{\left(1+\xi_{22}\eta_{22}V_{f}\right)}{\left(1-\eta_{22}V_{f}\right)}$$
(3)$$G_{12}=G_{m}\frac{\left(1+\xi_{12}\eta_{12}V_{f}\right)}{\left(1-\eta_{12}V_{f}\right)}$$
(4)$$\xi_{11}=2\left(l\times t\right)+40{\left(V_{f}\right)}^{10}$$
(5)$$\xi_{22}=2\left(w\times t\right)+40{\left(V_{f}\right)}^{10}$$
(6)$$\xi_{11}={\left(w\times t\right)}^{1.73}+40{\left(V_{f}\right)}^{10}$$
(7)$$\eta_{11}=\frac{\left(\frac{E_{f}}{E_{m}}-1\right)}{\left(\frac{E_{f}}{E_{m}}+\xi_{11}\right)}$$
(8)$$\eta_{22}=\frac{\left(\frac{E_{f}}{E_{m}}-1\right)}{\left(\frac{E_{f}}{E_{m}}+\xi_{22}\right)}$$
(9)$$\eta_{11}=\frac{\left(\frac{G_{f}}{G_{m}}-1\right)}{\left(\frac{G_{f}}{G_{m}}+\xi_{12}\right)}$$

Finite element analysis was performed to analyze the effects of UHMWPE fibers on the PP matrix and to analyze the behavior of the elastic properties. In the finite element analysis, the three-dimensional (3D) modeling of the composite material was constructed using the ANSYS Material Designer using the homogenization technique. A homogenization theory was developed to rigorously calculate the effective properties of the materials or composites with periodic arrangement; the calculation was performed by assuming that the structure was composed of a continuous connection of infinitely small unit cells [41,42,43]. In the case of finite element analysis through homogenization theory, it has symmetrical boundary conditions in all aspects. Therefore, a symmetrical boundary condition was given to each side. Based on this homogenization theory, the 3D model of the composite material was constructed, as shown in Figure 2. For the modeling, a hypothetical representative volume element (RVE) was formed and set to PP. The UHMWPE fiber simulated as a cylinder was dispersed inside the PP RVE. It was set as a basic modeling condition, such that the UHMWPE fibers oriented inside based on the set volume fraction were randomly arranged. The mesh type used for the finite element analysis was SOLID187, and the mesh size was set to 0.5 μm. Because SOLID187 can realize second-order displacements, it is suitable for forming meshes of irregular shapes. To minimize the property prediction sensitivity based on the mesh size, the sizes of the elements used in the PP matrix and UHMWPE fiber were set under the same conditions. All of the mesh sizes were analyzed under the same conditions to minimize uncertainty due to the mesh sizes. Because the length of the fibers increases with the aspect ratio of the UHMWPE fibers in the composite, the number of fibers was reduced to maintain a fixed volume fraction. Based on these modeling rules, the model was constructed by varying the volume fraction and aspect ratio of the UHMWPE fiber. Therefore, as the RVE size of the modeling varied according to the conditions, there were no fixed nodes and elements, and the number of nodes and elements varied for each composite model according to the aspect ratio and volume fraction of the UHMWPE fibers. Meanwhile, the Monte Carlo method was applied to the code used for creating the PP/UHMWPE composite material analysis model in order to obtain UHMWPE fibers and a PP matrix that were evenly distributed, in addition to non-overlapping UHMWPE fibers. The Monte Carlo method or Monte Carlo experiment involves an algorithm that mathematically approximates the value of a function via repeated random sampling. By adopting the Monte Carlo method, the UHMWPE fibers exhibit a constant stochastic distribution for each location on a PP matrix. Figure 2 shows the shape modeled based on the model generation theory used in the finite element analysis.

The boundary conditions for performing FEA are shown in Figure 3. To calculate the elastic properties according to the axial direction of each model, a force of 1 N was applied in the axial direction, and a displacement boundary condition was set on the opposite side. The reasons for applying a force of 1 N are as follows. As it is used to calculate the elastic modulus, a high force is not required. The elastic behavior was calculated by applying a low force to the composite material model. In addition, to calculate the shear modulus, as shown in Figure 3b, the shear modulus was calculated by applying a low force to the surface.

## 3. Results

Changes in elastic properties based on the aspect ratio of the PP/UHMWPE fibers were analyzed using finite element analysis and the Halpin–Tsai model. The elastic properties were predicted by varying the volume fraction of the UHMWPE fibers from 5% to 25% and varying the aspect ratio from 3 to 15 in the finite element analysis and Halpin–Tsai model. Figure 4 shows the calculated elastic properties. As the analysis results featured three variables (elastic properties, aspect ratio, and volume fraction), they are depicted in a 3D graph for clarity. The x-, y-, and z-axes represent the aspect ratio, volume fraction, and elastic properties, respectively. The results from the finite element analysis show that, when the volume fraction of the UHMWPE fiber was 5% and the aspect ratio was 3, the lowest modulus of elasticity was obtained, i.e., 1670.6 MPa. When the volume fraction of the UHMWPE fiber was 25% and the aspect ratio was 15, the highest elastic modulus value was obtained, i.e., 5902.5 MPa. Additionally, the results show that the elastic modulus in the fiber length direction (E_11_) was proportional to the aspect ratio and volume fraction. Meanwhile, calculations performed using the Halpin–Tsai model indicated the lowest and highest elastic modulus values of 1658.6 and 5457.4 MPa, respectively. A comparison of both methods shows that the results yielded by them differed more significantly when the aspect ratio and volume fraction were higher. The result of the finite element analysis shows that at an aspect ratio of 3 and volume fraction of 5%, the modulus of elasticity in the perpendicular direction of the fiber (E_22_) was 1502.8 MPa, and that the maximum modulus of elasticity obtained was 2407.6 MPa. The modulus of elasticity in the perpendicular direction of the UHMWPE fiber (E_22_) increased depending on the aspect ratio and volume fraction, although the magnitude of the increase was less than that of the modulus of elasticity in the longitudinal direction of the fiber (E_11_). However, the difference in the results yielded by the finite element analysis and the Halpin–Tsai model was insignificant. Figure 5 shows the calculated shear modulus (G_12_) of the composite material containing UHMWPE fibers. At a low aspect ratio and volume fraction, the shear modulus of the composite material yielded by the finite element analysis was 515.9 MPa. As the volume fraction increased, the shear modulus increased continuously; specifically, the shear modulus was 823.0 MPa when the volume fraction was 25%. In addition, as the volume fraction increased in the finite element analysis, the variability of the shear modulus increased with the aspect ratio. In contrast, the shear modulus calculated using the Halpin–Tsai model increased continuously with the volume fraction and aspect ratio.

A contour graph (see Figure 6) was constructed to derive the elastic modulus based on the volume fractions and aspect ratios used in the finite element analysis. A contour graph is used to represent a 3D curved surface on a 2D plane. In particular, when 3D points are represented by (x, y, z), for a specified z value, a curve connecting the points (x, y) that corresponds to this value is known as a contour line. Two methods can be used to draw contour lines, i.e., using a rectangular mesh constructed by the intersection of parallel grid lines, and using a triangle mesh constructed by connecting the diagonals of squares. Contour lines can be useful for expressing the data obtained via tests and measurements in two dimensions. As x and y are continuous values, a value exists between the lattices, which can be obtained easily via interpolation. Even if missing values occur in the data, they can be replaced by interpolation. To exploit the advantages of such interpolations, the result of the finite element analysis was analyzed based on the contour graph. First, to calculate the physical properties of the longitudinal elastic modulus (E_11_) of the PP/UHMWPE composite material reinforced with UHMWPE fiber, using Figure 6, the modulus of elasticity can be calculated based on the aspect ratio and volume fraction. Based on Figure 4 and Figure 6, the modulus of elasticity in the longitudinal direction increased with the aspect ratio and volume fraction of the UHMWPE fiber-reinforced composite material. The area denoted by ① in Figure 6 is the area where the result of the finite element analysis agrees well with the result of the Halpin–Tsai model. To predict the aspect ratio and volume fraction of the UHMWPE fibers in this area, the longitudinal elastic properties of the fibers can be used, which can be easily predicted using the Halpin–Tsai model. Figure 7 shows the change in the transverse elastic modulus (E_22_) of the UHMWPE fibers calculated via the finite element analysis. The results show that the modulus of elasticity in the direction perpendicular to the UHMWPE fiber increased proportionally to the aspect ratio, although it was not affected significantly by the volume fraction of the fiber. Additionally, the modulus of elasticity in the orthogonal direction of the fiber decreased slightly as the volume fraction increased. However, based on the Halpin–Tsai model, the orthogonal elastic modulus of the fiber was affected only by the aspect ratio of the UHMWPE fiber, i.e., the volume fraction did not exert any effect. As shown in region ① of Figure 7, the orthogonal elastic modulus of the fiber obtained via the finite element analysis and the Halpin–Tsai model matched well, regardless of the aspect ratio of the fiber until the volume fraction of the UHMWPE fiber was 13. However, as the volume fraction increased, the results of the Halpin–Tsai model deviated from those of the finite element analysis, as shown in region ② in Figure 7. A verification performed based on the shear modulus (G_12_) graph presented in Figure 8 confirmed that the shear modulus obtained via the finite element analysis increased with the volume fraction of the UHMWPE fiber. However, based on the Halpin–Tsai model, the shear modulus increased with the aspect ratio, whereas it was unaffected by the volume fraction. In addition, the deviation between the results of the finite element analysis and the Halpin–Tsai model was calculated to be 10% or higher. Hence, when predicting the physical properties of a composite material comprising UHMWPE fibers, at a low fiber volume fraction of 13 or less, one may assume that the elastic moduli obtained via finite element analysis and the Halpin–Tsai model agree well, whereas the shear modulus does not match well. Therefore, the analysis and the Halpin–Tsai model yielded similar results in terms of the effect of the cross-section of the UHMWPE fiber on the PP matrix, i.e., the elastic modulus in the longitudinal direction and the modulus in a perpendicular direction. However, the finite element analysis and the Halpin–Tsai model did not yield similar results for physical properties such as the shear modulus exerting on the surface of the composite material. Hence, the equations and finite element model must be supplemented by interpolation coefficients that are obtained experimentally.

## 4. Discussion

In general, the bonding strength of fiber-reinforced composite materials is affected the most significantly by the interfacial bonding strength between the fiber as the reinforcing agent and the matrix as the base material. The interface of the composite material transfers the externally applied impact energy, stress, and deformation from the matrix to the fibers. In addition, the effect of moisture can be suppressed by reducing the gap between the composite material and each of the fibers and the reinforcement. As such, the interface between the reinforcement and the matrix of the composite material is vital to the composite material. Because the composite material modeled via the finite element and numerical analyses in this study was imposed with boundary conditions that resulted in a good bonding strength at the interface, as shown in Figure 9, the modulus of elasticity in the fiber length direction increased with the aspect ratio and volume fraction of the UHMWPE fibers. If the volume fraction and aspect ratio of the UHMWPE fibers increase, then the x–y section of the 3D model will be modeled differently, as shown in Figure 9. When the UHMWPE fiber exhibited a volume fraction of 5% and an aspect ratio of 3, the interface density between the matrix and UHMWPE fiber was lower than that when its volume fraction was 25% and its aspect ratio was 15. The results show that the lengthwise modulus of elasticity of the fiber increased with the volume fraction of the fiber owing to the effect above and an increase in the aspect ratio of the fiber. In addition, the E_11_ calculated using the Halpin–Tsai model deviated slightly from the results of the finite element analysis at low aspect ratios. However, when the aspect ratio increased to 15, the results obtained via the finite element analysis and the Halpin–Tsai model deviated further. Therefore, the Halpin–Tsai model is applicable for predicting the modulus of elasticity at low aspect ratios.

Meanwhile, the transverse elastic modulus of the fiber was not significantly affected by the volume fraction and aspect ratio of the UHMWPE fiber, as shown in Figure 4. Based on Figure 10, this phenomenon occurred because the ratio of the cross-sectional area of the fiber was not affected significantly by the volume fraction and aspect ratio of the UHMWPE fiber. In addition, because the UHMWPE fiber was assumed to be cylindrical in this study, the width and thickness of the fiber were the same; hence, one may assume that it did not exert any significant effects. The number of cross-sectional areas of the UHMWPE fibers was different (as shown in Figure 10) because the number of UHMWPE fibers decreased as the aspect ratio increased, while the volume fraction of the UHMWPE fibers remained constant. However, based on Figure 10, the variables affecting the elastic modulus E_22_ in the *y*-axis direction were the height and width of the fiber cross-section; hence, they did not significantly affect the E_22_ of the composite material. In fact, the E_22_ obtained via the finite element analysis and the Halpin–Tsai model was almost identical. Therefore, the Halpin–Tsai model can be used to predict composite materials, regardless of the aspect ratio and volume fraction.

Additionally, the behavior of the shear modulus (G_12_) of the composite material shown in Figure 5 is associated with the characteristics of the composite material modeled (as shown in Figure 2). The 3D model of the composite material used in this study was modeled using the homogenization technique, which can infinitely expand the RVE in which the UHMWPE fibers are randomly placed. Therefore, using the aspect ratio of the randomly placed UHMWPE fibers inside the matrix will not yield the consistently predicted values of the shear modulus. Meanwhile, the shear modulus (G_12_) calculated using the Halpin–Tsai model was proportional to the volume fraction and aspect ratio of the fiber, because the fibers were assumed to be oriented in a lattice shape. To develop a PP/UHMWPE composite material based on the graphs presented in Figure 6, Figure 7 and Figure 8, it is concluded that the aspect ratio and volume fraction of the UHMWPE fibers can be determined based on the target properties of the composite material.

## 5. Conclusions

In this study, to develop a PP/UHMWPE composite material, the effect of the volume fraction and aspect ratio of the UHMWPE fibers inside a PP matrix was comparatively analyzed via finite element analysis and the Halpin–Tsai model. The elastic properties of the composite material predicted using the Halpin–Tsai model and the results calculated via the finite element analysis agreed well with each other at low aspect ratios and volume fractions. However, as the volume fraction and aspect ratio of the UHMWPE fibers increased, the deviation between the results obtained via the two abovementioned approaches increased. This occurred because the UHMWPE fibers were randomly configured in the composite 3D model used for the simulation, whereas the composite material assumed in the Halpin–Tsai model was calculated based on the lattice and regular orientation. As the aspect ratio increased, the effect of the orientation of the UHMWPE fibers at the interface increased when the matrix material and the fibers were combined in the longitudinal direction inside the composite material. Meanwhile, the orthogonal elastic modulus (E_22_) obtained via the two abovementioned approaches agreed well with each other. This was attributable to the cross section of the UHMWPE fiber, i.e., because the cross section was assumed to be circular, the results yielded by the two analysis methods were similar. In addition, based on the Halpin–Tsai model, the shear modulus (G_12_) of the composite materials increased proportionally with the volume fraction and aspect ratio of the fibers. However, the results of finite element analysis showed that the shear modulus increased with the volume fraction of the UHMWPE fibers, whereas it did not increase as the aspect ratio increased.

The properties of the composite material were calculated via the finite element analysis by comprehensively analyzing these trends. By referring to a contour graph, the elastic properties can be predicted based on the volume fraction and aspect ratio when developing PP/UHMWPE composite materials. In addition, the properties of the added UHMWPE fibers can be predicted based on the physical properties of the material to which those fibers were added. In addition, the conclusions that can be obtained through the results of this study are as follows. When developing PP/UHMWPE composite materials, it can be used as a database for detailed composite material design by considering the volume fraction and aspect ratio of UHMWPE fibers depending on the required performance of the components used and the composite material that meets demand. In the future, there is a need to discuss these parts with experimental data to produce quantitative results. In addition, when a composite material is constructed using a short-, long-, or continuous-fiber reinforcement mechanism, its properties should be investigated both experimentally and analytically prior to its manufacture.

## Figures and Tables

**Figure 1 polymers-14-04851-f001:**
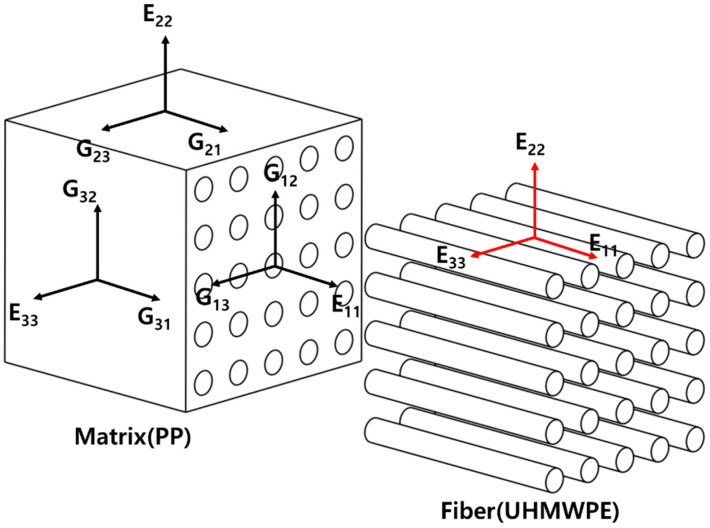
Tensor notation of fiber-reinforced composite materials.

**Figure 2 polymers-14-04851-f002:**
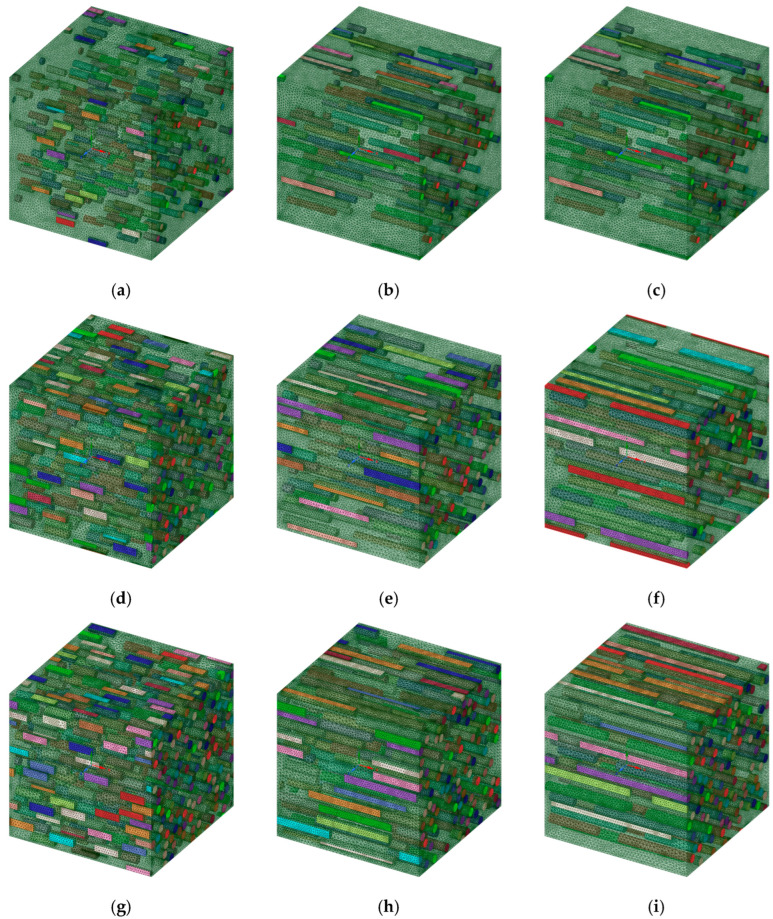
Composite material finite element analysis 3D model, (**a**) VF5%—Aspect ratio 3, (**b**) VF5%—Aspect ratio 9, (**c**) VF5%—Aspect ratio 15, (**d**) VF15%—Aspect ratio 3, (**e**) VF15%—Aspect ratio 9, (**f**) VF15%—Aspect ratio 15, (**g**) VF25%—Aspect ratio 3, (**h**) VF25%—Aspect ratio 9, (**i**) VF25%—Aspect ratio 15.

**Figure 3 polymers-14-04851-f003:**
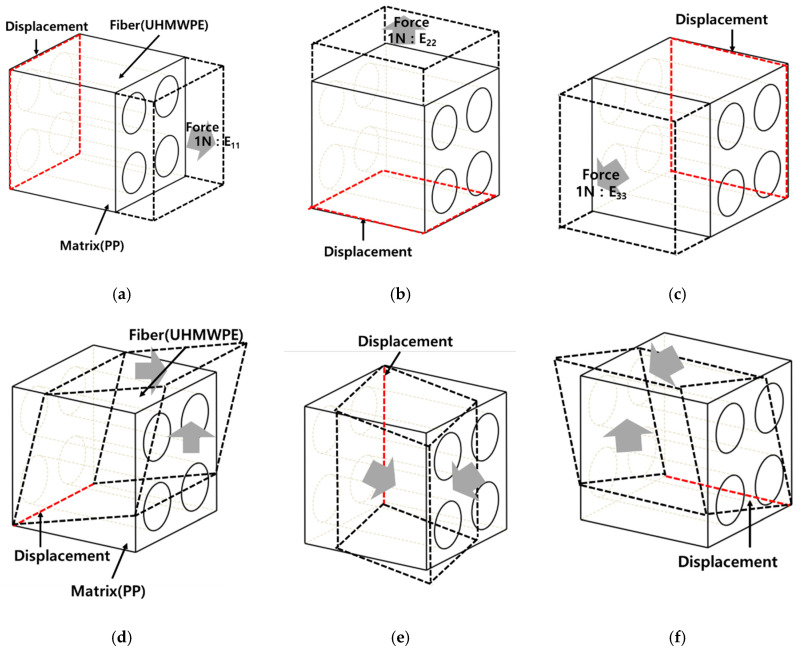
Boundary conditions for FEA. (**a**) E_11_ Young’s modulus, (**b**) E_22_ Young’s modulus, (**c**) E_33_ Young’s modulus, (**d**) G_12_ Shear modulus, (**e**) G_13_ Shear modulus, (**f**) G_23_ Shear modulus.

**Figure 4 polymers-14-04851-f004:**
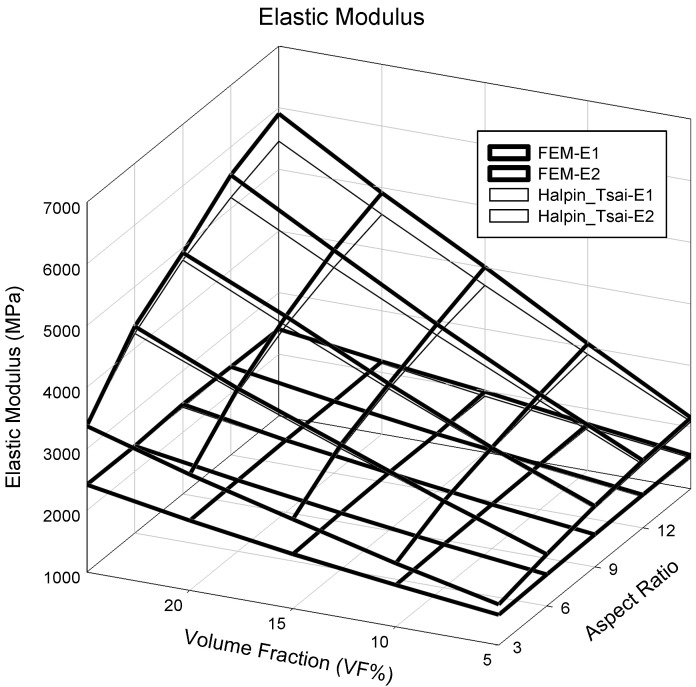
Elastic modulus result based on the volume fraction and aspect ratio.

**Figure 5 polymers-14-04851-f005:**
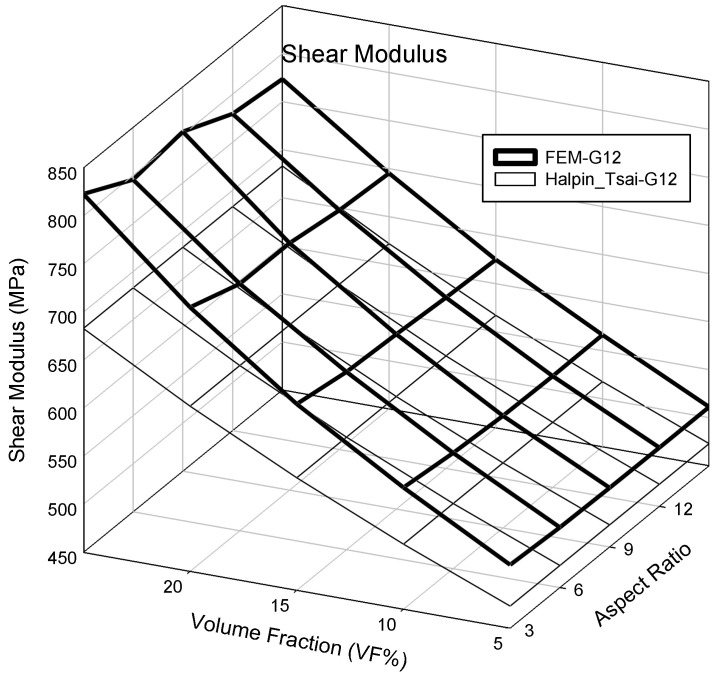
Shear modulus result based on volume fraction and aspect ratio.

**Figure 6 polymers-14-04851-f006:**
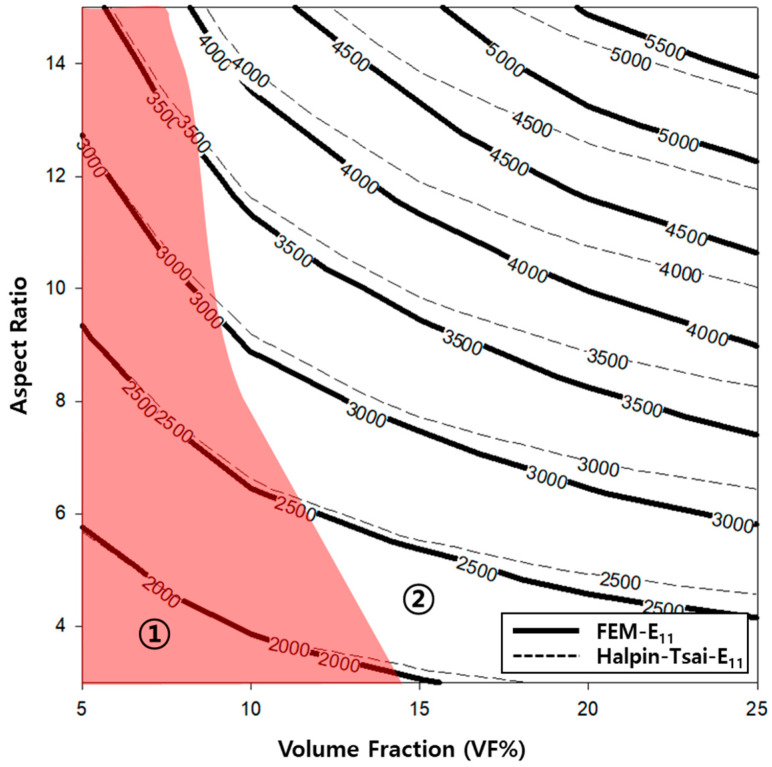
Contour graph showing elastic modulus (E_11_) as functions of the volume fraction and aspect ratio.

**Figure 7 polymers-14-04851-f007:**
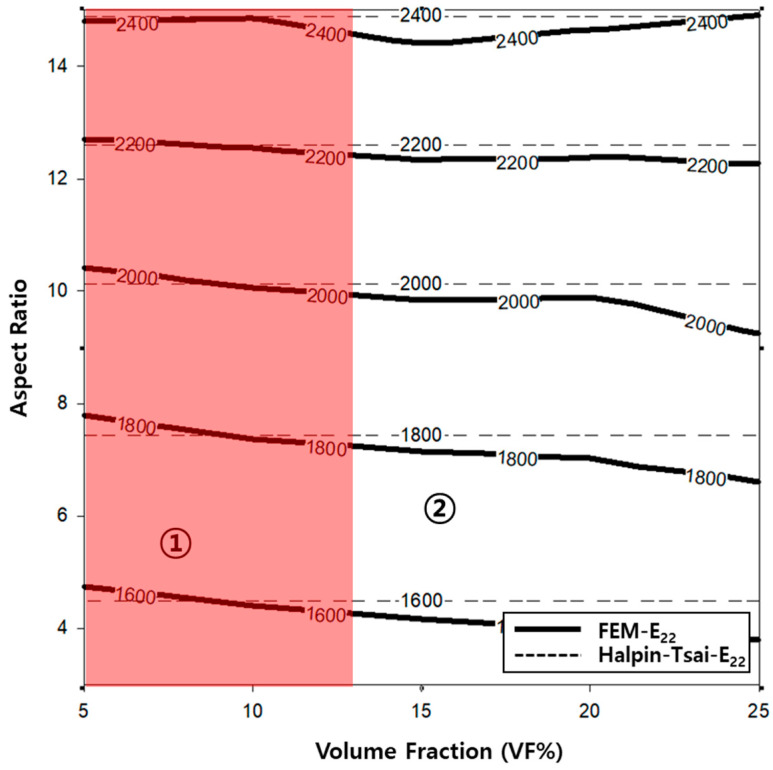
Contour graph showing elastic modulus (E_22_) as functions of the volume fraction and aspect ratio.

**Figure 8 polymers-14-04851-f008:**
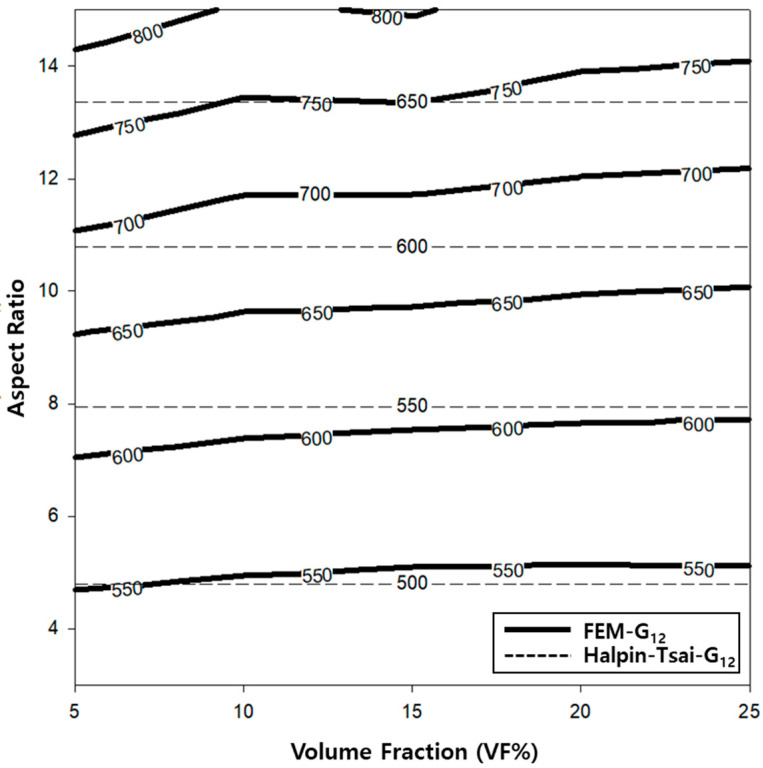
Contour graph showing shear modulus (G_12_) as functions of the volume fraction and aspect ratio.

**Figure 9 polymers-14-04851-f009:**
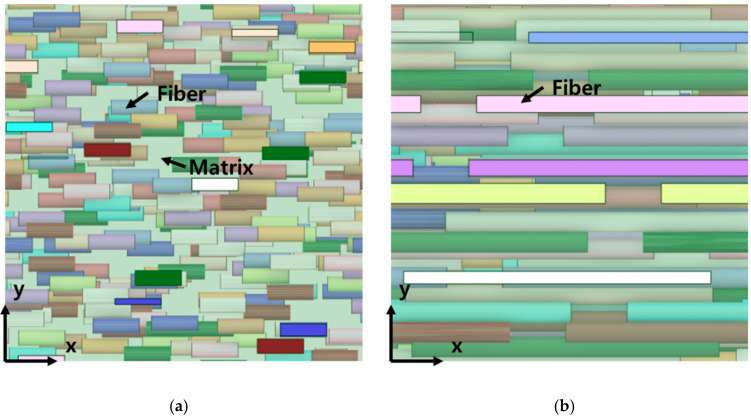
x–y cross section of 3D model based on the volume fraction and aspect ratio, (**a**) VF5%—Aspect ratio 3, (**b**) VF25%—Aspect ratio 15.

**Figure 10 polymers-14-04851-f010:**
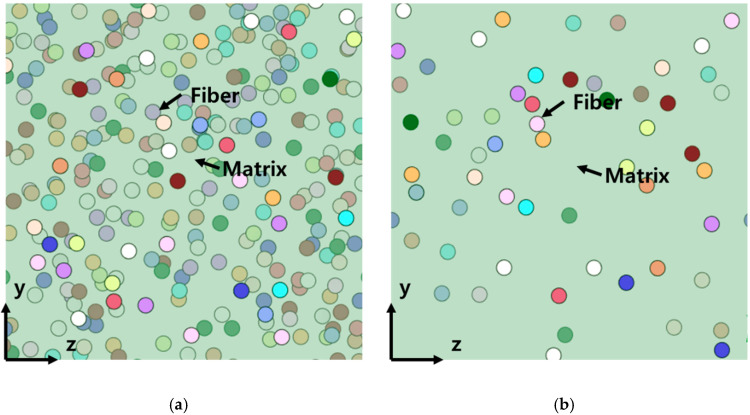
x–y cross section of the 3D model based on the volume fraction and aspect ratio, (**a**) VF5%—Aspect ratio 3, (**b**) VF5%—Aspect ratio 15.

**Table 1 polymers-14-04851-t001:** Properties of PP and UHMWPE used in the current study.

	PP	UHMWPE
modulus of elasticity (MPa)	1325	25,000
shear modulus (MPa)	432.29	10,417
Poisson’s ratio	0.43	0.20
bulk modulus of elasticity (MPa)	3154.8	13,889.0
density (kg/m^3^)	904	950

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
