# Peer review of "Numerical Investigation of the Elastic Properties of Polypropylene/Ultra High Molecular Weight Polyethylene Fiber inside a Composite Material Based on Its Aspect Ratio and Volume Fraction"

_polymers, 2022, doi:10.3390/polym14224851_

Round 1

Reviewer 1 Report

This manuscript studies the elastic properties of polypropylene/ultrahigh molecular weight polyethylene fiber composites based on a finite element analysis model. The manuscript does not deserve the publication in its current format; however, since the subject is interesting, the authors are allowed to make a major revision before final decision.

1. Please avoid using abbreviations in the title.

2. Title needs revision. It is mandatory to include the polypropylene in the title because it is the matrix. My recommendation is “Numerical Investigation of Elastic Properties of Polypropylene/Ultrahigh Molecular Weight Polyethylene Fiber Composites Based on Aspect Ratio and Volume Fraction”.

3. It is highly recommended to avoid using the lumped references such as [3-10], [11-14], [15-21], [22-28], [29-32] and etc. without any discussion. Each sentence is allowed to have one or at maximum, two references. Each reference should be cited with relevant discussion or justification.

4. One of the distinguishing features of academic writing is that it is informed by what is already known, what work has been done before, and/or what ideas and models have already been developed. Thus, in academic texts, writers frequently make reference to other studies and to the work of other authors. It is important that writers guide their readers through this literature. It is the purpose of the literature review section of a paper or to show the reader, in a systematic way, what is already known about the research topic as a whole, and to outline the key ideas and theories that help us to understand this. As well as being systematic, the review should be evaluative and critical of the studies or ideas which are relevant to the current work. The Introduction does not have an appropriate literature survey.

5. The objectives, contributions and novelties of the study should be highlighted in the last paragraph of the Introduction. Which gaps of the literature have been covered by the study? This issue should be addressed in a separate paragraph as the last paragraph of the introduction.

6. Please provide the explanations for Eq. (1)-(9). They have been presented without any information and explanation.

7. Please provide supporting references for Eq. (1)-(9).

8. Is the information of Table 1 measured by the authors? If so, please provide the methodology. If not, please provide supporting references for this data.

9. It is better to combine sections three and four in a section entitled Results and Discussion. However, I leave it to you.

10. The finite element method utilized in this study needs a validation by comparing the results with an experimental or theoretical study.

11. Conclusions need a major revision. It is long in the present form. Conclusions are shorter sections of academic texts which usually serve two functions. The first is to summarize and bring together the main areas covered in the writing, which might be called ‘looking back’; and the second is to give a final comment or judgement on this. The final comment may also include making suggestions for improvement and speculating on future directions.

Author Response

Thank you for your review.

I worked hard to improve it.

Thank you.

Reviewer 2 Report

This research reports on a numerical study on elastic Properties of UHMWPE fiber incorporated into composite materials based on aspect ratio and volume fraction.

This is a topic that can potentially provide useful information in the area of composite materials and structures and their applications to various engineering structures such as plates and shells. However, there are important issues that need to be addressed by the authors through "Major Revision" in the interest of improving this work before I can recommend this manuscript for publication.

My comments/recommendations are as follows:

1) The authors need to clearly and explicitly elaborate on the novelty and significant contribution of their work. To my knowledge, the numerical investigation into the elastic properties of fiber reinforced composite materials using aspect ratio and volume fraction has been broadly studied and disseminated within the literature. 

2) Aspect ratio in composite materials is a very general term which refers to the ratio associated with the length-to-diameter. Such a ratio can vary significantly depending on the type of fiber used. Continuous fibers have long aspect ratios, while discontinuous fibers have short aspect ratios. Continuous-fiber composites normally have a preferred orientation, while discontinuous fibers generally have a random orientation. You haven't taken into account his important factor in your numerical analysis. Is your study a limited case-study? 

3) The reviewer doesn't understand what the point of using the term "ultrahigh molecular weight polyethylene (UHMWPE)" is in this research? It is just a fiber name and doesn't contribute to numerical modeling at all. On the other hand, UHMWPE is not required in numerical modeling. Each fiber has its own unique elastic properties which is a fact and common sense. 

4) The authors have asserted that they have used Finite Element Analysis (FEA) to calculate the elastic properties of properties of the PP/UHMWPE composite materials. The reviewer doesn't agree with the authors. There is no numerical investigation using FEA in the present manuscript. The reviewer can't see any FEA simulation procedures and results. For example:

-How were the loading and the material properties introduced into the FE software?

-How were the boundary conditions prescribed to your FE model? It is of extreme importance. The boundary conditions must be shown on the FE model.

-What type of element has been selected from the material library? exp: shell elements if plane stress is considered for thin structures or solid elements for thick and/or moderately thick structures.

-Any mesh refinement?

-Any result validation?

None has been conducted which is a serious flaw in the eye of the reviewer.

5) Another important issue is the Finite Element (FE) mesh refinement/convergence study. Thus, the current description of the commercial numerical apparatus used for this purpose is extremely poor. To rectify the current situation, the following additions are necessary:

(a) Please conduct mesh sensitivity analysis for qualitative verification of your FE simulation results. Mesh sensitivity analysis is conducted for convergence studies and/or adaptive analysis to determine and control the numerical errors, respectively.

(b) Give a complete characteristic of the applied commercial elements in respect to the applied loading, the number of element nodes and the number and kind of degrees of freedom per each node.

(c) For each numerical calculation, the number of elements and degrees of freedom must be given.

(d) You must discuss the issue of potential sources of modeling and discretization errors in your numerical analysis; also, the methods of determination and controlling such errors should be mentioned.

6) The literature review isn't adequate. The authors are apparently not much familiar with some of the leading research regarding the analytical and FEA simulation of composite materials and structures under different loading conditions. The introduction is currently not concise and informative. The author must discuss the following references in the introduction portion of the revised manuscript and elaborate on the novelty of their work with respect to the studies below:

"Localized failure analysis of internally pressurized laminated ellipsoidal woven GFRP composite domes: Analytical, numerical, and experimental studies", Archives of Civil and Mechanical Engineering, Vol:19, pp:1235-1250.  

"A semi-empirical approach to evaluate the effect of constituent materials on mechanical strengths of GFRP mortar pipes", Structures, Volume 36, February 2022, Pages 493-510.

7) The manuscript requires significant language revision. I strongly recommend the authors have their paper reviewed by a native English speaker to improve its written expression.

I believe responding to the above-mentioned comments and revising the manuscript accordingly are essential before considering the paper for publication.   

Author Response

(The authors gave the same response as above.)

Round 2

Reviewer 1 Report

This manuscript studies the elastic properties of polypropylene/ultrahigh molecular weight polyethylene fiber composites based on a finite element analysis model. The reviewer believes that the authors made a desirable revision on the manuscript and the revised version can be considered for publication if the editor and other reviewers agree.

Reviewer 2 Report

After carefully reviewing the manuscript, I came to the conclusion that the authors have adequately responded to my comments and revised the manuscript. This version of the manuscript is an improvement. As such, I believe the revised manuscript meets the necessary requirements for publication in Polymers.